# Evaluation of Causal Inference Models to Access Heterogeneous Treatment Effect

## Abstract

Causal inference has gained popularity over the last years due to the ability to see through correlation and find causal relationships between covariates. There are a number of methods that were created to this end, but there are lacking studies that do a systematic benchmark between those methods, including the benefits and drawbacks of using each one of them. Moreover, most methods use IHDP or Jobs data sets to show that their method is better, but there are some concerns that using these data sets could lead to biased conclusions that do not reflect the real data scenario. This research compares a number of those methods on how well they assess the heterogeneous treatment effect using synthetically created data sets that are representative of real data. These synthetically created data sets are generated using flexible generative models that provide data sets that are representative of real data. We compare the error between those methods and discuss in which setting and premises each method is better suited.

## 1 Introduction

Over the last decades, causal inference has begun to gain importance within machine learning and data science discussions. Literature on the topic has grown considerably, especially after Judea Pearl came up with the do-calculus framework (Pearl, 1995), formalizing the theory of causality.

With the increasing growth of causal inference methods, it is becoming necessary to compare those methods and create a benchmark. Other authors have previously made surveys or comparisons of the most well known available methods for causal inference (Yao et al., 2021) (Shalit et al., 2017) (Alaa & Schaar, 2018), being only the theory behind each method or alongside with comparison when ran with synthetically or semi-synthetically created data sets. Most papers presenting causal inference methods and in surveys rely on the frequently used data sets (namely IHDP (Hill, 2011) and NSW's jobs data set (LaLonde, 1986)) to compare such methods with the baseline. Using these data sets as ground truth for what the performance of the models would look like could lead to a biased conclusion, as it may not necessarily reflect properties of real data (Curth et al., 2021).

We present a causal inference model benchmark using data sets created using the RealCause data generation process (Neal et al., 2020), which would better address the issues presented in the currently used data sets. In order to do the benchmark and further develop causal inference research and applications to the community, we developed an open source Python library in which all models in the benchmark were implemented, as well as additional features that are useful for causal inference research.

For this article, we will be using the Newman-Rubin framework of potential outcomes for causal inference, described in Section 2. Section 3 explains the models that are going to be used for the benchmark. Section 4 will approach the methodology, data sets used for the benchmark and metrics used. Sections 5 and 6 will approach the results of the benchmark, along with a discussion on how the models performed in each setting. Finally we will conclude in section 7, along with future work.

## 2 Neyman-Rubin Causal Model

In this study we will use the Neyman-Rubin framework for potential outcomes (Sekhon, 2008). The framework consists of a non-parametric model where each unit has two potential outcomes, one if the unit was treated and another if it was not. The treatment effect is calculated as the difference between such two potential outcomes. However, only one of the outcomes can be observed in practice, as the unit was either treated or not, which creates a conundrum called "the fundamental problem of causal inference".

Let $Y_i(1)$ be the potential outcome if the unit were to receive the treatment and $Y_i(0)$ otherwise. The individual treatment effect (ITE) is defined as:

$$\tau_i = Y_i(1) - Y_i(0) \tag{1}$$

Because $Y_i(1)$ and $Y_i(0)$ are never observed simultaneously, it is considered a missing data problem. Despite not being able to observe the ITE, we can make inferences about the average treatment effect (ATE). If the problem was designed as a randomized controlled trial, potential outcomes $Y_i(0), Y_i(1)$ would be independent of the treatment assignment T:

$$E[Yi(j)|T = 0] = E[Yi(j)|T = 1], j \in \{0, 1\} \tag{2}$$

This means that the group of the treated unit and the group of the untreated are comparable. Another way to see this is that the two groups are exchangeable, i.e., if the treated and the untreated were swapped it would return the same results. That is called the ignorability or the exchangeability assumption. The ATE would be defined as:

$$\tau = E[Y_i(1)|T = 1] - E[Y_i(0)|T = 0] = E[Y|T = 1] - E[Y|T = 0] \tag{3}$$

For an observational experiment, the independence assumption may not hold (as it is normally the case). Some assumptions are necessary in this case to get proper treatment effects, as described below.

### 2.1 Conditional exchangeability/unconfoundedness

In the cases where we have non-random treatment assignments (for instance, older people might be more susceptible to receive treatment than younger folks in certain clinical trials), we cannot use the ignorability assumption and thus the ATE formula. The unconfoundness assumption is an extension of the ignorability assumption, where $Y_i(1)$, $Y_i(0)$ are independent of treatment T, given features X:

$$(Y(0), Y(1)) \perp\!\!\!\perp T | X \tag{4}$$

That means that there are no unobserved features that are confounders of the treatment assignment. Therefore, the ATE can be calculated as:

$$\tau = E_x[E[Y|T = 1, X] - E[Y|T = 0, X]] \tag{5}$$

### 2.2 Positivity

The positivity assumption determines that, in all subgroups, there is a probability greater than zero and lower than one to receive the treatment. If there is a subgroup that either received only treatment or control assignments, it would be not possible to calculate the treatment effect for that group.

$$0 < P(T = 1|X) < 1 \tag{6}$$

With this assumption, it is possible to evaluate equation 5 without dealing with undefined elements. As a side note, as the number of features grows, the likelihood of violating this assumption increases accordingly.

## 2.3 Stable unit-treatment value assumption (SUTVA)

This assumption deals with two ideas, the first being no interference: the potential outcome from one unit is not affected by the treatment assignment other units received. The second idea is no hidden variance: if a unit were to receive treatment, the outcome would be $Y = Y(1)$. That means that there are no possible different outcomes stemming from the same treatment.

## 3 Data Generation Process

The easiest way to evaluate a causal inference model is using synthetic data that simulates the covariates $X$, the treatment $T$ and the outcome $Y$. This gives access to the true outcome mechanism $P(Y|T,X)$ and the treatment selection mechanism $P(T|X)$. With it, we can access the true average treatment effect by $\tau = E_x[E[Y|T = 1, X] - E[Y|T = 0, X]]$.

Using this method can lead to several disadvantages. The first one is that there is no guarantee that the synthetically created data is similar to data presented in the real world, making it difficult to evaluate how the models are going to perform in a real scenario. The second is that, depending on how the data is created, can lead to properties that make some models perform better than other by exploiting it.

An improved way is to generate data in a semi-synthetic fashion, where the covariates $X$ are pulled from real data, so $P(X)$ is realistic, but still simulating $P(T|X)$ and $P(Y|T,X)$. That can lead to the advantage that $P(X)$ is drawn from real data, so it is closer to being comparable to real life situations, but $P(T|X)$ and $P(Y|T,X)$ are still unrealistic. This is where most datasets used for comparison (including IHDP) falls into.

To avoid the forementioned problems, the RealCause (Neal et al., 2020) library was used to create the data sets. A library that simulates real data using data generating process (DGP) that are indistinguishable from real world data. First, it fits generative models $P_{model}(T|X)$ and $P_{model}(Y|X,T)$ that matches $P(T|X)$ and $P(Y|X,T)$. The idea is to sample $X$ from the real data and then sample $T$ from $P_{model}(T|X)$ and finally $Y$ from $P_{model}(Y|X,T)$.

The architecture of the generative models follow the structure of TARNet (Shalit et al., 2017) to learn two conditionals $P_{model}(Y|X,T = 0)$ and $P_{model}(Y|X,T = 0)$, that way, encoding the importance of $T$ in the architecture. It uses MLP to get a function $h(X)$ from the variables $X$. It also uses three separate MLP of the same architecture to model $T$, $Y|T = 0$ and $Y|T = 1$ using $h(X)$ as input.

## 4 Models

In this benchmark, we analyzed 8 algorithms as causal models, as described on Table 1. We investigated models based on linear regression, matching algorithm with nearest neighbors, models based on propensity score, tree-based model and metalearners. The tree-based model and metalearners are further discussed in the next section.

### 4.1 Randomized trees ensemble embedding

The randomized trees ensemble setting is inspired by the causal forest framework (Wager & Athey, 2018). The key concept for this setting is using a forest algorithm to model the relation between the covariates (without the treatment flag) and the outcome, and then matching treatment and control samples in the forest's latent embedding space. This setting is expected to overcome issues regarding confounder selection and heterogeneous treatment effects.

ExtraTrees models were chosen as the forest algorithms, because such models are expected to perform well in high dimensional settings with noisy covariates. The configuration is shown in Figure 1. In this benchmark,

| Model Category | Name | Description |
|---|---|---|
| Propensity | Inverse Propensity of Treatment Weighting(IPTW) | Average treatment effect is computed as a weighted average with the same propensity scores used in the PSM setting described above |
| Matching | Propensity Score Matching | Propensity score is modeled using logistic regression with balanced class weights; matching is performed on such score by a k-nearest neighbors model with 10 neighbors |
| Matching | Nearest Neighbors | K-nearest neighbor models with 10 neighbors |
| Linear | Linear Regression | Multiple linear regression without regularization for data sets with continuous targets |
| Linear | Logistic Regression | Logistic regression with balanced class weights for data sets with binary targets |
| Tree-based | Randomized trees ensemble embedding | ExtraTrees embedding of the covariates followed by k-nearest neighbors matching with 10 neighbors |
| Metalearner | X-Learner | Metalearner using xgboost as base model |
| Metalearner | RA-Learner | Metalearner using xgboost as base model |
| Metalearner | DR-Learner | Metalearner using xgboost as base model |

ExtraTrees with both default parameters and optimized parameters were included. As mentioned above, ExtraTrees models were trained to fit the outcome variable using the covariates. To make the models honest, as described in (Wager & Athey, 2018), we then proceed to use this model to encode the validation data set and then train two k-nearest neighbor models in the latent embedding space, one for the control set and another for the treatment set. Finally, counterfactuals and evaluation are computed for the test set.

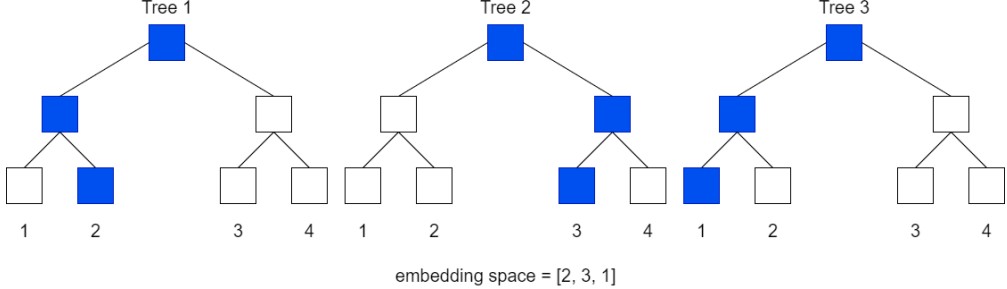

Figure 1: Forest-model embedding. The trained model yields an encoded representation of the training data on its leaves, forming an embedding which can be used in further steps of causal inference.

## 4.2 X-learner

The benchmark also included X-learners with the base model being a XGBboost model. Both models with the default parameters and with the hyperparameter tuning step were analyzed, as was the case for the randomized trees ensemble setting. The X-learner is implemented as described in (Künzel et al., 2019). The first step is to create two models (using the base model), one using only the data from the treated ($\mu_1$) and another using only the control data ($\mu_0$):

$$\mu_1 = E[Y(1)|X_{treated}] \tag{7}$$

$$\mu_0 = E[Y(0)|X_{control}] \tag{8}$$

The second step is to calculate the treatment effect from the treated and the treatment effect from the control:

$$D_1 = y_1 - \mu_0(X_{treated}) \tag{9}$$

$$D_0 = \mu_1(X_{control}) - y_0 \tag{10}$$

The third step is to create the models $\tau_0$ and $\tau_1$ based on the results from the last step:

$$\tau_0 = E[D_0|X_{control}] \tag{11}$$

$$\tau_1 = E[D_1|X_{treated}] \tag{12}$$

The last step to get the treatment effect is weighting the resulted models:

$$\tau(x) = g(x) * \tau_0(x) + (1 - g(x)) * \tau_1(x), g(x) \in [0, 1] \tag{13}$$

The function $g(x)$ represents the propensity score, which is calculated as the output of a logistic regression trained using all data set.

### 4.3 RA-Learner

The regression-adjusted learner, proposed in (Curth & van der Schaar, 2021), create the two base models ($\mu_0$ and $\mu_1$) using the same method as X-Learner. After calculate the pseudo-outcome of the treatment effect using the following formula:

$$\tilde{Y}_{RA} = T(Y - \mu_0(X)) + (1 - T)(\mu_1(X) - Y) \tag{14}$$

The last step is to create a model using the pseudo-outcome as target given X.

### 4.4 DR-Learner

The doubly-robust learner, proposed in (Kennedy, 2020) uses the same idea as the RA-Learner, creating two base model $\mu_0$ and $\mu_1$. They calculate the pseudo-outcome based on the AIPW estimator:

$$\tilde{Y}_{DR} = (\frac{1}{\pi(X)} - \frac{1-T}{1-\pi(X)})Y + [(1 - \frac{T}{\pi(X)})\mu_1(X) - (1 - \frac{1-T}{1-\pi(X)})\mu_0(X)] \tag{15}$$

where $\pi(X)$ is the propensity score ($E[T|X]$). This pseudo-outcome has the nice property of being unbiased if either the propensity score or the first step regressions is correctly specified. The last part is the same as previous models, creating a model that directly output the pseudo-outcome from features X.

## 5 Methodology

The main goal of this work is to benchmark different models for causal inference in a variety of settings, without prior knowledge of the data structure or the relation between the features, the treatment, and the target variables. We used models that represent different kinds of approaches: propensity score matching, IPTW, linear regression, k-nearest neighbors, extratrees embeddings (representing the matching algorithms), x-learner, ra-learner, dr-learner.

The metalearners approach (x-learner, ra-learner and dr-learner experiments) were divided into two different settings: one using the default hyperparameters of the models and another in which we did a step of hyper optimization to select the best parameters to use for the models. The hyperparameter search used was a random search of 60 iterations on each step separately with the following settings:

- max_depth: [3, 5, 7, 9],

- learning_rate: stats.loguniform(0.001, 0.6),

- subsample: [0.5, 0.6, 0.7, 0.8, 0.9, 1.0],

- colsample_bytree: [0.4, 0.5, 0.6, 0.7, 0.8, 0.9, 1.0],

- colsample_bylevel: [0.4, 0.5, 0.6, 0.7, 0.8, 0.9, 1.0],

- min_child_weight: [0.5, 1.0, 3.0, 5.0, 7.0, 10.0],

- gamma: [0, 0.25, 0.5, 1.0],

- reg_lambda: [0.1, 1.0, 5.0, 10.0, 50.0, 100.0],

- n_estimators: stats.randint(10, 300)

Each model in the analysis was trained in every data set with standardized features whenever required by the model. We computed the precision in estimation of heterogeneous effect (PEHE) of the predicted average treatment effect (ATE) and grouped by the data generating processes (DGPs). The results are displayed in Tables 1 and 2.

### 5.1 Data Sets

The data sets used in this study are created using the RealCause library, which uses flexible generative models to provide data sets that yield ground-truth values of the causal effects and are representative of real data (Neal et al., 2020). Two different settings were used to create the data sets, each one using three different data sets as data generation input (lalonde CPS, lalonde PSID and Twins). For each pair of setting / data generation input, 100 different data sets were created using different premises:

1. using default parameters (the same settings used in (Neal et al., 2020))

2. in a setting with small treatment effect (using treatment_scale parameter at 0.01)

All data sets used the same random seeds as the RealCause article, being able to reproduce the results easily.

### 5.2 Metrics and results evaluation

The main metric analyzed in these data sets is the precision in estimation of heterogeneous effect (PEHE) of the predicted average treatment effect (ATE). This metric was chosen as it is standard among other benchmarks and causal models evaluations. It is scored as:

$$PEHE = \sqrt{\frac{1}{N} \sum_{i=1}^{N} (\tau(X_i)^* - \tau(X_i))^2} \tag{16}$$

where $\tau(X_i)^*$ is the estimated treatment effect and $\tau(X_i)$ the given treatment effect.

The data sets were analyzed in 2 different settings:

1. Using the default parameters of the GDP;

2. In a setting that the treatment effect is scaled by a factor of 0.01

Table 1: 95% confidence interval for the regular setting (green shows best results)

| | lalonde_cps | lalonde_psdi | twins |
|---|---|---|---|
| linear | (12615.831 12713.273) | (23677.98 23948.855) | (0.498 0.5) |
| iptw | (12089.008 15034.042) | (18804.116 19136.529) | (0.498 0.5) |
| knn | (8672.187 8748.106) | (13781.035 14038.855) | (0.504 0.506) |
| psm | (9100.666 9196.501) | (13535.361 13803.272) | (0.504 0.505) |
| causal_forest | (9432.527 9628.414) | (13985.838 14239.201) | (0.516 0.518) |
| x_learner | (12131.62 13029.21) | (20133.031 23894.167) | (0.467 0.469) |
| x_learner_hyper | (11842.742 12213.129) | (20498.623 21574.619) | (0.479 0.484) |
| ra_learner | (9000.458 9319.077) | (13882.284 17288.867) | (0.445 0.446) |
| ra_learner_hyper | (9074.053 9257.702) | (13156.077 13534.49) | (0.475 0.482) |
| dr_learner | (9002.755 9320.287) | (13876.307 17283.068) | (0.438 0.439) |
| dr_learner_hyper | (11553.868 13816.387) | (16235.63 21298.582) | (0.486 0.502) |

### 5.3 Benchmark implementation

The pseudocode shown below implements the benchmark logic. It consists of a nested loop through all data sets and models, which are selected depending on the target type (regression models if the target is binary and classification models if the target is continuous).

Evaluation metrics are computed and stored after fitting the model, and the process is repeated for each data set.

```
for dataset in datasets:
        if dataset target is continuous:
                models = regression models
        else if dataset target is binary:
                models = classification models

        for model in models:
                fit model using data set
                compute evaluation metrics
                store evaluation metrics
```

## 6 Results

In this section, benchmark results are presented as 95% confidence intervals for the precision of estimation heterogeneous effect (PEHE) of predicted ITE and ATE error for each group of data generation model and setting. The tables show the results for each setting (regular and with scaled treatment effect).

## 7 Discussion

Benchmark results clearly show that no model is the winner for all settings, along with other findings. The main takeaways are described below.

- RA-learners and DR-learners were the best performers across the board. It is probably due to the fact that such models are more robust to noisy covariates that might introduce bias to the ATE estimation. The exception is the DR-Learner with hyperparameter optimization, which performed worse than only using the default parameters;

- Propensity score matching (PSM) and nearest-neighbors performed best on the Lalonde's datasets, which is probably due to it being more robust to confounding and heterogeneous treatment effects;

Table 2: 95% confidence interval for the scaled setting (green shows best results)

| | lalonde_cps | lalonde_psdi | twins |
|---|---|---|---|
| linear | (612.085 617.041) | (1963.883 1993.195) | (0.072 0.074) |
| iptw | (586.775 728.727) | (1560.763 1590.861) | (0.072 0.074) |
| knn | (420.665 424.669) | (1142.641 1168.336) | (0.073 0.075) |
| psm | (441.459 446.438) | (1122.426 1148.438) | (0.072 0.075) |
| causal_forest | (457.495 467.147) | (1159.675 1185.87) | (0.074 0.076) |
| x_learner | (590.858 637.15) | (1685.482 1994.271) | (0.067 0.069) |
| x_learner_hyper | (572.153 591.649) | (1700.254 1770.955) | (0.07 0.072) |
| ra_learner | (436.07 451.853) | (1155.789 1437.928) | (0.064 0.066) |
| ra_learner_hyper | (438.771 447.719) | (1095.343 1134.727) | (0.068 0.071) |
| dr_learner | (436.039 451.899) | (1142.791 1437.266) | (0.063 0.065) |
| dr_learner_hyper | (559.593 652.806) | (1372.05 1724.793) | (0.068 0.072) |

- Linear regression performed worst, probably due to the non-linear nature of the data generation process;

- Matching algorithms performed more consistently, with narrower 95% confidence;

- IPTW did worse than PSM, probably because when propensity scores are not well modeled (due to lack of correlation between covariates and the treatment flag), simply matching samples by score tends to do better than using such scores as inverse weights for ATE estimation;

- Hyper optimization did not systematically improve ATE estimation for any of the meta-learners, which indicates that a better prediction power in the target does not necessarily imply in better causal inference, as considering confounding yields better treatment effect predictions than finding correlation;

- The performance rank from models using the regular setting and the scaled setting did not change, which indicates that the models are not sensible to smaller treatment effect.

Simpler matching models like nearest neighbors and propensity score matching are good starting points to benchmark results, but it is important to address the premises from the data generation process first. Does the data not violate the positivity assumption? Do we have a hint that the target has linear interactions with the features? Then a linear regression could be a good starting point. If it is believed that the target has a complex iteration with the features, maybe matching algorithms are better choices. If the data has a very strong heterogeneous treatment effect, meta learners like the ra-learner and dr-learner can be good solutions.

## 8 Conclusion

There is no silver bullet that is going to perform best in all settings. X-learners have gotten the best results overall, but were not the best across all DGPs. Linear regression and propensity score matching have performed well across the settings, even though they are considered to be simpler in comparison to the other methods analyzed. Also, hyper optimization has not been a game changer for ATE estimation. The combination of all these factors seems to indicate that a better prediction power in the outcome does not necessarily imply in better causal inference.

Also, no confounder selection was performed, which might have hindered the performance of models that are more prone to issues with noise and biases. Including confounder selection principles is an important next step to further investigate causal model performance, because accounting for confounding is more effective than modeling correlation between covariates and outcomes in order to improve prediction of treatment effects.

Further research must be addressed to get a better understanding of the models, including settings where there is poor overlap.

We also hope that the open source library made available can be useful in further developing the area of causal inference, not only in research but also in real-life applications, with support of the community to continue the evolution of the library.

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
