# OpenReview forum: "Evaluation of Causal Inference Models to Access Heterogeneous Treatment Effect"
_TMLR — Rejected by TMLR_

### Review · Reviewer_2dok · 2023-11-06

**Summary Of Contributions:**

This is a small experimental comparison of some popular methods for heterogeneous treatment effect estimation. It is directly based on the RealCause library, using 100 different data sets for the 3 different settings respectively. The methods are compared only based on accuracy.

**Audience:**

No

**Claims And Evidence:**

No

**Requested Changes:**

Times (5 times maybe) more work should be done and the paper should be fully rewritten.
I do not believe the additional work could be done in the rebuttal timeline, and, after the additional work and rewriting, this should be actually another paper.
So, I do not provide a detailed request list, but please refer to the Weaknesses for suggestions.

**Strengths And Weaknesses:**

### Strengths
Maybe, the re-implementation of the methods in Python is useful for some people.

### Weaknesses
**The work does not "better address the issues presented in the currently used data sets"** as described in the paper as no "exhaustive algorithm comparison", no comparisons on "a range of different data generation processes", and no comparisons of "the advantages and the restrictions of using each method".

Specific points are as below.

I am not sure what the author(s) mean by "algorithm(ic) comparison". I assume that should be something systematic and automatic, but there are no such things in the paper.

With linear models excluded, only 7 methods are compared. So the comparison is far from "exhaustive".

The categorization of the methods into "Baseline models" and "State-of-the-art models" is not helpful and confusing. A more useful categorization is "balancing methods", "regression methods", and "Mixed (double) methods" as in [1, Sec 4]

[1]Cousineau, Martin, et al. "Estimating causal effects with optimization-based methods: A review and empirical comparison." European Journal of Operational Research 304.2 (2023): 367-380.

I believe each setting in the RealCause has a same causal graph, if so, this is far from "a range of different data generation processes".

Accurary is the only kind of metrics considered in the comparison. We cannot discuss "the advantages and the restrictions of using each method" to a satisfactory extent by this. In fact, in Sec 6 Discussion, there are only some obvious observations reading from the Table.

I do not understand why we'd like to consider the "small treatment effect" setting. In fact, the last point in Discussion says this does not matter, and this is obvious for me.

**The writing is very sloppy and confusing**, for example:

What is the "Logistic regression with balanced class weights" in Table 1? This is not even mentioned outside of the Table.

There is the sentence "The results are displayed in Tables 1, 2 and 3.", but Table 1 is not results.

"In Table 2, we grouped the low dimensional data sets by modification used to generate the samples." What is the "modification"? Do you mean the 3 settings?

Do the "regular setting" and "scaled setting" in the Table 2 & 3 refer to default or small treatment effect scale? This is confusing because you did not use exactly the same wording.

At the end of Sec 4.2, you say "4 different settings", but there are only 3 in the list. And, 1 & 3 in the list is already mentioned in Sec 4.1, while 2 low overlap is totally not dealed with in the experiments?

At the beginning of Sec 4.3, you mention "regression models" and "classification models" but there are no definitions.

At the beginning of Sec 5, you say there are confidence intervals for PEHE and ATE error, but there is only one interval for a method?

---

### Review · Reviewer_iFfw · 2023-11-07

**Summary Of Contributions:**

This paper proposes a causal inference model benchmark using datasets created by the RealCause data generation process. This benchmark addresses several issues of previous benchmarks, such as only using synthetic datasets, and single data generation process. They also develop a Python library to facilitate future causal inference research.

**Audience:**

Yes

**Claims And Evidence:**

Yes

**Requested Changes:**

Please address the Weaknesses. I may be willing to recommend for acceptance if they are addressed well.

**Strengths And Weaknesses:**

Strengths
- This paper makes a relatively thorough benchmark and summary for different causal inference methods given different settings and datasets. The proposed Python library is a effective tool for future research.

Weaknesses
- The wrtting needs to be further improved. There are some unidiomatic expressions and typos.
- The Section 2 is a little superfluous. For causal inference researchers, most of them are familiar with Neyman-Rubin frameworkm, and thorough introduction is not necessary.
- For the SOTA methods in Table 1, the latest one is proposed in 2021. I think newer methods should be discussed here.
- An introduction to the RealCause is needed to help readers understand your work more easily.
- In Section 4.3, is the regression model and the classification model confused? In my understanding, the binary target needs the classification model, instead of the regression model.

---

### Review · Reviewer_rGDx · 2023-12-01

**Summary Of Contributions:**

The paper aims to perform an empirical study of causal inference methods for estimating heterogeneous treatment effects. The authors aims to particularly conduct this comparative study under different data regimes and analyse the limitations and strength of each model. The study is motivated by the need, in the authors perspective, to evaluate the plethora of treatment effect estimation method on more "realistic" data generating processes. The main contributions are:

1. Use the RealCause data generation approach proposed by Neal et al 2020 to generate data where the true treatment effect can be obtained. Explore different settings of the parameters compared to Neal et al 2020
2. With the generated datasets, run different treatment effect estimation methods based on matching, reweighting and meta-learners and tree ensemble methods and evaluate their performance in terms of the PEHE (precision in estimation of heterogenous effect)

**Audience:**

No

**Broader Impact Concerns:**

The paper does not present any particular ethical implications.

**Claims And Evidence:**

No

**Requested Changes:**

Please refer to the weaknesses section above.

**Strengths And Weaknesses:**

**Strengths**:
* The paper makes a good effort in motivating the study and in thoroughly explaining some of the methods used for estimating treatment effects. The methodology is straightforward and was clearly explained.
* The authors make a good effort in open sharing a python library with implementation of the methods explored in the paper as well as the data generating process.

**Weaknesses**

* Motivation: \
The paper aims to empirically evaluate treatment effect estimation methods. As acknowledged by the authors, such studies have been extensively done in the literature. The paper aims to motivate their work by introducing a differentiation in the datasets used. \
Indeed, a major limitations in evaluating treatment effect methods using observational data is the lack of ground truth where the true effect is known. While this is a know limitations, datasets based on randomized control trials as well as semi-synthetic datasets are available and used in the literature. \
However, the authors rely of the RealCause process proposed by Neal et al 2020 to generate what they deem "more realistic" datasets. In the RealCause process, generative models are used to produce synthetic and semi-synthetic data with better representativeness of the real data and with known ground-truth causal effects. \
Given the main novelty of the paper resides in this use of different datasets and discarding previous similar work based on this factor, the motivation should go further than solely taking as given the claims of the RealCause reference.

* Empirical evaluation: \
Other than the motivation, the main goal of the paper is to empirically evaluate these methods under different data generating processes and extensively study the conditions under which some methods work and their limitations. However, in the paper, the empirical evaluation is severely lacking. Indeed, a lot of effort and space was used to introduce standard definitions and assumptions in causal inference as well as some of the methods studies namely meta learners and randomized tree method. While this helped in the clarity of the paper, it came at the expense of a more detailed experimental design and evaluation. \ For a purely empirical study work, I would have appreciated a more extensive experimentation, performing hyperparameter search, doing ablation studies, sensitivity analysis, etc. However, the method were compared solely on one metric and a few different settings. \

* Analysis: The analysis of the results is also lacking. This can be attributed to the scarceness of experimental results to compare (see point above on the empirical evaluation). While their was an attempt to explain the results, the explanation is mostly based on assumptions or hypothesis without further evaluation. I would have appreciated if the authors went further and tested some of the possible explanation they postulate for the results. This would have better informed the conclusions on which methods work best under which setting.

* Writing style: at times the writing seems too informal and subjective. This is particularly true for the introduction.

---

### Decision · Action_Editor_Ak9t · 2024-01-01

**Recommendation:** Reject

**Comment:**

This paper compares different causal inference methods on a synthetic data which have groundtruth. The paper reviews some existing causal inference methods and compare them on the synthetic dataset. As pointed out by all reviewers, this paper does not make a scientific contribution to the causal inference area. The empirical study's limitations are evident in its restricted dataset scope and the insufficient details regarding experimental design and discussion of results. Consequently, due to these shortcomings, my recommendation is to reject this submission.

**Audience:**

No, the work does not provide meaningful new findings or methodology.

**Claims And Evidence:**

No. The results are limited to the synthetic data used in the paper. The limited empirical results are not sufficient to derive a conclusion on which causal inference method should be preferred than the others.